# Bone Marrow Fibrosis at Diagnosis and during the Course of Disease Is Associated with TP53 Mutations and Adverse Prognosis in Primary Myelodysplastic Syndrome

**DOI:** 10.3390/cancers14122984

**Published:** 2022-06-16

**Authors:** Youshan Zhao, Juan Guo, Sida Zhao, Roujia Wang, Lei Shi, Ying Fang, Zheng Zhang, Luxi Song, Dong Wu, Chunkang Chang

**Affiliations:** Department of Hematology, Shanghai Jiao Tong University Affiliated Sixth People’s Hospital, Shanghai 200000, China; zhaoyoushan@sjtu.edu.cn (Y.Z.); guojuan0321@sjtu.edu.cn (J.G.); star.zhao@alu.fudan.edu.cn (S.Z.); 19921219@sjtu.edu.cn (R.W.); angelsights@alumni.sjtu.edu.cn (L.S.); elainefong@sjtu.edu.cn (Y.F.); david33211@sjtu.edu.cn (Z.Z.); songluxi@139.com (L.S.)

**Keywords:** myelodysplastic syndrome, bone marrow fibrosis, TP53, prognosis

## Abstract

**Simple Summary:**

To understand the role of bone marrow fibrosis and its molecular changes in myelodysplastic syndrome, we retrospectively analyzed data from 814 patients. Older age, lower hemoglobin, unfavorable karyotype and higher BM blast were more often observed in patients with moderate/severe fibrosis. Cases with bone marrow fibrosis had reduced overall survival. TP53, U2AF1 and KMT2D mutations were more frequent in patients with moderate/severe fibrosis. In addition, 15.1% of patients progressed to moderate/severe fibrosis during the follow-up interval. The clinical features, mutation landscape and prognosis of patients with progressed fibrosis were similar to those patients with moderate/severe fibrosis at diagnosis. We concluded that bone marrow fibrosis was associated with reduced overall survival in primary MDS and correlated with TP53 mutations both at the time of initial diagnosis and during the course of the disease.

**Abstract:**

The prognostic significance of bone marrow fibrosis (MF) grade in patients with myelodysplastic syndrome (MDS) is still debated and the molecular changes remain unclear. In our large cohort, a normal reticulum was found in 211 (25.9%) patients, whereas MF1, MF2 and MF3 were detected in 478 (58.7%), 90 (11.1%) and 35 (4.3%) patients at initial diagnosis, respectively. Patients with MF often correlated with some poor prognostic characteristics, including older age, anemia, unfavorable karyotype, higher BM blast and a higher IPSS-R category. For the entire cohort, the median OS was not reached, 30, 16 and 15 months for patients with MF 0, 1, 2 and 3, respectively. After adjusting for IPSS-R, the hazard ratio for mortality was 1.56 (95% CI, 1.18–2.06) for patients with MF1, 2.29 (95% CI, 1.61–3.27) for patients with MF2 and 2.75 (95% CI, 1.69–4.49) for patients with MF3 compared with those with MF0. The mutational landscape of 370 patients showed that TP53, U2AF1 and KMT2D mutations were more frequent in patients with MF2-3. In addition, of the 408 patients with MF0-1, 62 patients (15.1%) progressed to MF2-3 during the follow-up interval. The clinical features, mutation landscape and prognosis of patients with progressed fibrosis were similar to those of patients with MF2-3 at diagnosis. We concluded that BM fibrosis (MF1, 2 and 3) was an adverse prognosis feature in primary MDS and correlated with TP53 mutations both at the time of initial diagnosis and during the course of the disease. Therefore, BM fibrosis should be included in the revised prognostic scoring system and carefully considered in treatment selection.

## 1. Introduction

Myelodysplastic syndromes (MDS) are clonal hematopoietic disorders with very distinct natural histories [1]. Bone marrow fibrosis (BMF), defined as increased deposition of reticulin in the bone marrow, can be found in 10–20% of patients with primary MDS and had been suggested to be a poor prognostic factor in MDS [2,3,4]. According to the European bone marrow fibrosis network criteria, BMF was graded as MF 0, 1, 2 and 3 [5]. Although a number of prognostic scoring systems, such as the Revised International Prognostic Scoring System (IPSS-R), have been established to risk-stratify patients with MDS for treatment selection, BM fibrosis is not covered. The absence of BMF as an additive factor in IPSS-R could relate to the low number of patients assessed for this feature (19%) as well as the variable ways the degree of fibrosis was reported from the different institutions in the study [6]. Its distinctly clinical features and prognostic role in MDS need to be confirmed in larger sample size studies.

Research on hematological neoplasms has advanced greatly with the application of next-generation sequencing techniques. Activating mutations in JAK2, CALR or MPL genes is a prominent feature of primary BM fibrosis, but these mutations are infrequent in MDS [7,8]. Several studies have clarified the detailed mechanism of high-frequency mutations such as SRSF2, SF3B1 and U2AF1 in the pathogenesis of MDS [9,10,11]. Genetic mutation profiles of MDS with fibrosis are beginning to be studied [8,12]. Systematic analysis of the molecular changes in MDS with BM fibrosis is needed.

In the current study, by retrospective analysis of a large cohort of patients, we designed the research to confirm the malignant characteristics, inferior outcomes and mutation landscape in primary MDS with BM fibrosis at initial diagnosis. Moreover, BMF developing over the course of disease in patients with MDS was evaluated.

## 2. Patients and Methods

### 2.1. Patient Samples

A total of 814 primary MDS patients diagnosed in Shanghai Jiao Tong University Affiliated Sixth People’s Hospital between September 2009 and November 2019 were analyzed retrospectively. The diagnostic criteria were referenced from the World Health Organization (WHO) in 2016 [13]. The prognostic risk stratifications were performed according to the IPSS-R [6]. Patients with history of myeloproliferative neoplasms or myelodysplastic/myeloproliferative syndrome were excluded. Median follow-up duration was 48 months. All cases had an appreciable bone marrow biopsy at the time of diagnosis. Informed consent was obtained from all subjects according to institutional review board–approved protocols, which were carried out in accordance with the Declaration of Helsinki.

### 2.2. BM Fibrosis Assessment

Bone marrow aspirates with biopsy were evaluated in patients at different points along disease diagnosis and course. Reticulin stains were performed on all BM biopsy specimens. BM fibrosis was graded according to the European Bone Marrow Fibrosis Network (EUMNET) criteria as MF 0, 1, 2 and 3 [5]. In cases with heterogeneous reticulin deposition, fibrosis was assessed based on the more prevalent pattern. The bone marrow fibrosis was graded blinded to the clinical features of the presentation. For patients with MF 2 or 3, bone marrow biopsy was reevaluated by other hematopathologists to determine the degree of BM fibrosis. Patients were divided into three groups: MF0, MF1 and MF2-3, according to their different overall survival.

### 2.3. Mutational Analysis

Next-generation sequencing was performed on 370 patients as routine standard testing since January 2016 in our department. BM DNA was of adequate quality. The testing was detected by KingMed Diagnostics (Guangzhou, China), a clinical certified laboratory using the Illumina platform (Illumina, San Diego, CA, USA). A 29-gene panel was carried out as previously reported [14]. All assays were performed blinded to the study end points, by pathologists who were not involved in patient management.

### 2.4. Statistical Methods

Statistical analyses were conducted with SPSS software, version 17.0 (SPSS, Inc.). Categorical variables were compared using Fisher’s exact test or an χ2 test as appropriate. Survival curves were prepared via the Kaplan–Meier method and compared via the log-rank test. Multivariate Cox proportional hazards models were used to calculate hazard ratios with 95% confidence intervals of association pertaining to the relationship between risk factors and survival. All reported *p* values were 2-sided and considered significant at 0.05.

## 3. Results

### 3.1. MDS Patients with BM Fibrosis Emerged Adverse Clinical Feature

A total of 814 patients with primary MDS were enrolled, with a median age of 59 years (range from 17 to 91). The study population consisted of 471 male and 343 female cases. At initial diagnosis, a normal reticulum (MF0) was found in 211 (25.9%) patients, whereas mild (MF1), moderate (MF2) and severe (MF3) BM fibrosis was detected in 478 (58.7%), 90 (11.1%) and 35 (4.3%) patients, respectively. The 125 (15.4%) patients with substantial bone marrow fibrosis (MF2-3) were compared with those with MF0 and MF1. The main clinical and baseline characteristics of patients are summarized in Table 1. In brief, patients with MF [2,3] had an older age than those with MF0 or MF1. The MDS-EB1/2 type occurred more often in patients with MF1 (31.6%) and MF2-3 (44%) when compared with patients with MF0 (15.2%, all *p* < 0.001, Figure 1A). Analyzing the complete blood count data, cases with MF2-3 had a comparable leukocyte count and platelets but a lower hemoglobin level (72.6 g/L, *p* < 0.001) compared with patients with MF0 and MF1. In addition, patients with MF1 and MF2-3 often had a higher BM blast compared with those with MF0 (all *p* < 0.001, Figure 1B). According to IPSS-R, the poor/very poor karyotypes were observed more frequently in patients with MF1 and MF2-3 than those without fibrosis (all *p* < 0.001, Figure 1C). Moreover, patients with MF1 and MF2-3 often had a high and very high IPSS-R score compared with those with MF0 (Figure 1D). Based on these results, the presence of BM fibrosis, especially MF2-3, in MDS indicated a more aggressive feature.

### 3.2. BM Fibrosis Independently Affect Overall Survival

Median follow-up duration was 48 months (range 1–150 months) and 53.2% of the patients died. The median OS was not reached, 30 (95% CI, 24.7–35.3), 16 (95%CI, 10.5–21.5) and 15 (95% CI, 8.2–21.8) months for patients with MF 0, 1, 2 and 3 (*p* < 0.001, Figure 2A), respectively. The rate of transformation to acute myeloid leukemia was 19%, 24% and 20% for patients with MF1, MF2 and MF3, respectively, which is significantly higher than patients with MF0 (8.1%, *p* = 0.001). The prognostic significance of BM fibrosis on a series subset of patients was also concerned. For the subset of patients with MDS-EB-1/2, cases with MF2-3 only had a median OS of 11 months (95% CI, 8.6–13.4), which is significantly worse than those with MF0 (median OS 19 months (95% CI, 9.3–28.7)) or MF1 (median OS 17 months (95% CI, 14.3–19.7)) (Figure 2B). For the 424 patients with MDS-MLD, the median OS was not reached, 47 (95% CI, 38.4–55.6) and 18 (95% CI, 9.5–26.5) months for patients with MF0, 1 and 2–3 (*p* < 0.001, Figure 1C), respectively. In addition, the adverse survival of BM fibrosis was demonstrated in different IPSS-R score groups (Figure 2D–F). To further confirm the independent prognostic of different-grade MF in MDS, we employed multivariable analyses. After adjusting for the IPSS-R risk group, the hazard ratio (HR) for mortality was 1.56 (95% CI, 1.18–2.06, *p* = 0.002) for patients with MF1, 2.29 (95% CI, 1.61–3.27, *p* < 0.001) for patients with MF2 and 2.75 (95% CI, 1.69–4.49, *p* < 0.001) for patients with MF3 compared to those with MF0.

### 3.3. BM Fibrosis Associated with Gene Mutations of TP53, U2AF1 and KMT2D

Twenty-nine genes recurrently mutated in MDS were sequenced. Of the 370 cases included in this analysis, 127 (34.3%) were determined to be MF0, 199 (53.8%) MF1 and 44 (11.9%) MF2-3. The spectrum of gene mutations in all patients is shown in Figure 3A and Table 2. The most frequently mutated genes were TET2 (17.3%), ASXL1 (11.1%), DNMT3A (11.1%), SETBP1 (11.1%) and U2AF1 (10.5%), followed by ANKRD11 (9.7%), SF3B1 (9.5%), TP53 (8.9%), RUNX1 (8.6%), MPL (8.4%), JAK2 (7.8%), KIF20B (7.6%), IDH1/2 (7.0%), ROBO1 (6.2%) and BCOR (5.7%). Interestingly, a strikingly high prevalence of TP53 mutations (11/44, 27.3%) was found in patients with MF2-3 as compared with those patients with MF0 (6/127, 4.7%, *p* < 0.001) or MF1 (15/199, 7.5%, *p* = 0.001, Figure 3B). KMT2D mutations were seen to be more frequent in patients with MF2-3 than patients with MF1 (9.1% vs. 2.0%, *p* = 0.038). Moreover, U2AF1 mutations were detected in 9/44 (20.5%) patients with MF2-3 vs. 17/199 (8.5%) patients with MF1 (*p* = 0.03) and 13/127 (10.2%) patients with MF0 (*p* = 0.114). There was no significant difference in the rate of occurrence of JAK2 mutation across the different groups. CALR mutations, as well as other gene mutations (NRAS, KRAS, GATA2, ZRSR2, etc.) were detected at a low frequency and did not show any significant differences between patients with BM fibrosis and patients without it.

### 3.4. Bone Marrow Fibrosis Progression during the Course of Myelodysplastic Syndromes

To confirm the dynamic changes in BM fibrosis in MDS, we focused on patients with MF0 or MF1 at the time of diagnosis. Of them, 408 patients had follow-up bone marrow biopsy specimens available for assessment. The median bone marrow specimen number was 4 (range from 2 to 22 times). A total of 41 patients (10.0%) progressed to MF2 and 21 patients (5.1%) progressed to MF3 during the follow-up interval. Their clinical features, compared with patients with stable MF0-1 are shown in Table 3. Bone marrow fibrosis progression occurred more often in patients with MDS-EB-1/2, higher BM blast level, poorer karyotype and higher IPSS-R score. Of these fibrosis progressive patients, 56 were initially diagnosed as MF1 and 6 as MF0. Fibrosis progression rate was significantly higher in patients with MF1 than in patients with MF0 (56/317, 17.7% vs. 6/91 6.6%, *p* = 0.005). The median time for MF development was 11 months (range, 2–96 months). The median survival after bone marrow fibrosis developed was 9 months (range, 1–54 months). At the time of fibrosis progression, 20 patients had concomitant disease progression (based on the elevation of bone marrow blast cells), including 7 patients who developed leukemia.

In this cohort of patients, 170 patients had the mutation data at the time of diagnosis, including 28 cases with progressed MF and 142 patients with stable MF0-1. We further compared the differences in mutations between the two groups. As shown in Table 4, patients with progressed MF emerged with higher-frequency TP53 mutations than patients with stable MF0-1 (17.9% vs. 2.1%, *p* = 0.003). Moreover, RUNX1 mutations were seen to be more frequent in patients with progressed fibrosis than those patients with stable fibrosis (17.9% vs. 5.3%, *p* = 0.058).

The median overall survival of the 62 patients with aggravated BM fibrosis, similar to patients with MF2-3 at diagnosis, was significantly lower than those patients who had no fibrosis development (20 months vs. 79 months) (Figure 4A). In addition, the median LFS of patients with fibrosis progression was significantly worse than those patients with stable fibrosis (50 months vs. not reached, *p* < 0.001, Figure 4B).

## 4. Discussion

In the current study, we confirmed that the presence of BM fibrosis, not only MF2-3, but also MF1, was associated with more aggressive disease and unfavorable prognosis in patients with MDS. The gene mutation profile exhibited that TP53, U2AF1 and KMT2D mutations were correlated with BM fibrosis. What is more, we found that developed BM fibrosis in MDS patients also indicated poor clinical outcomes.

Our study indicated that BM fibrosis was correlated with some poor prognostic characteristics, including older age, anemia, unfavorable karyotype, higher BM blast and a higher IPSS category. These findings agreed with most of the previous studies [3,4,15,16], suggesting our study cohort was not selected and represented a common MDS population. These multiple correlations indicated that BM fibrosis characterized a distinct subgroup of MDS patients with more aggressive behavior. Therefore, more attention should be paid to the evaluation of bone marrow fibrosis at the time of initial diagnosis and decision making for treatment.

The prognostic significance of bone marrow fibrosis grade in MDS is still debated. Most previous studies have found that patients with moderate to severe reticulin fibrosis (MF2-3) often had a poorer overall survival compared with patients with MF0-1. So only MF2-3 was considered as MDS-F in studies [8,15,17,18]. More recently, Melody et al. [19] reported that only MF3 affected survival, when patients with MF3 were compared with those with MF0-2 fibrosis, based on an MDS database of 2624 patients. In their study, only 4% of patients were affected by BM fibrosis. Considering the large sample size, we also separately analyzed the impact of each grade of BM fibrosis on patient’s outcomes. The median OS was not reached, 30, 16 and 15 months for patients with MF0, 1, 2 and 3, respectively. The hazard ratio (HR) for mortality was 1.56, 2.29 and 2.75 for patients with MF1, 2 and 3 compared with those with MF0. Therefore, not only MF2-3, but also MF1 affected the outcomes of MDS in our study. The poor prognosis of patients with MF1 was consistent with the clinical features. As shown in Table 1, cases with MF1 often had more aggressive features than patients with MF0, such as a higher proportion of MDS-EB-1/2, poorer karyotype and higher IPSS-R score. Of course, these adverse characteristics were more pronounced in patients with MF2-3. In addition, the high rate of fibrosis progression in patients with MF1 contributed to the poor prognosis. This finding requires confirmation in other larger patient series.

Molecular changes observed in MDS with fibrosis remain to be fully unraveled. Ramos et al., found a high prevalence of cohesin complex mutations in patients with MDS with fibrosis as compared with those patients without it in a small sample study [8]. Loghavi et al., studied 67 de novo MDS with BM fibrosis and reported a high frequency of p53 overexpression and TP53 mutations in patients with MF2-3 [12]. Melody et al., noted that TP53 and SETBP1 variants are more frequently detected in patients with MF3. Our research reinforced this conclusion. TP53 mutations were detected in 27.3% patients with MF2-3 in our cohort, which was significantly higher than that in MF0 (4.7%). Additionally, even more interestingly, we further found that TP53 mutation was strongly associated with the progression of BM fibrosis in MDS. This evidence confirms that TP53 mutations are closely related to BM fibrosis in MDS. It is well-known that TP53 mutations in AML and MDS are usually associated with resistance to conventional therapeutics [20,21] and have been described to be predictive for a favorable response to decitabine, a hypomethylating agent, which decreases mutated p53 levels [22,23]. It has been suggested that MDS patients with BM fibrosis and TP53 mutations may respond well to decitabine combination treatment. This notion was in keeping with the study reported by Talha Badar et al., that 82% of advanced primary MF benefited from decitabine therapy [24]. However, Melody and colleagues considered that hypomethylating agents may have no obvious advantage in MDS with BM fibrosis. They found that the overall response was 39% among patients with MF3 compared to 35% among patients with MF0-2 who were treated with hypomethylating agents [19].

Gene mutations in epigenetic regulators (ASXL1, TET2, DNMT3A and EZH2) and RNA splicing (U2AF1 and SRSF2) were considered as the additional subclonal mutations and cooperated with the MPN driver mutation (JAK2, MPL or CALR) to play a key role in the pathogenesis of primary myelofibrosis [25,26,27]. SRSF2, ASXL1 and U2AF1 mutations predict inferior survival in primary myelofibrosis, independently of other risk factors [27]. Wang et al., reported that U2AF1 variants were more common in MDS or MDS-AML patients with MF2-3 [28]. In the current study, U2AF1 mutations accounted for 20.5% of patients with MF2-3. Moreover, KMT2D, responsible for the H3K4me1 modification found at enhancers, was mutated in 9.1% patients with MF2-3, higher than those with MF1 (2.0%). As epigenetic regulators, KMT2D mutations are common in lymphoma. Both gene mutations may represent myelofibrotic characteristics of MDS. Further research is needed to clarify their role in MDS fibrosis.

Study of BM fibrosis progression was rare in MDS in previous studies. Hu et al., identified 17 MDS patients with developed BM fibrosis during the course of the disease [29]. Eight of the seventeen patients showed increased bone marrow blasts, and five patients showed clonal cytogenetic evolution. In this study, only 20 of the 62 patients with fibrosis development had disease progression. The median OS and LFS of patients with aggravated BM fibrosis was significantly worse than those patients who had no fibrosis development. TP53 mutations were also associated with fibrosis progression in our cohort. The clinical features, mutation landscape and prognosis of patients with progressed fibrosis were similar to those patients with MF2-3 at initial diagnosis. Therefore, we believe that fibrosis progression is not an inherent manifestation of disease progression and has prognostic value.

## 5. Conclusions

Overall, MF1 as well as MF2-3, in MDS patients indicated adverse clinical features and poor outcomes at initial diagnosis and during the course of disease. Genes mutated in TP53, U2AF1 and KMT2D were correlated with fibrosis in MDS. BM fibrosis requires close attention throughout the course of the disease.

## Figures and Tables

**Figure 1 cancers-14-02984-f001:**
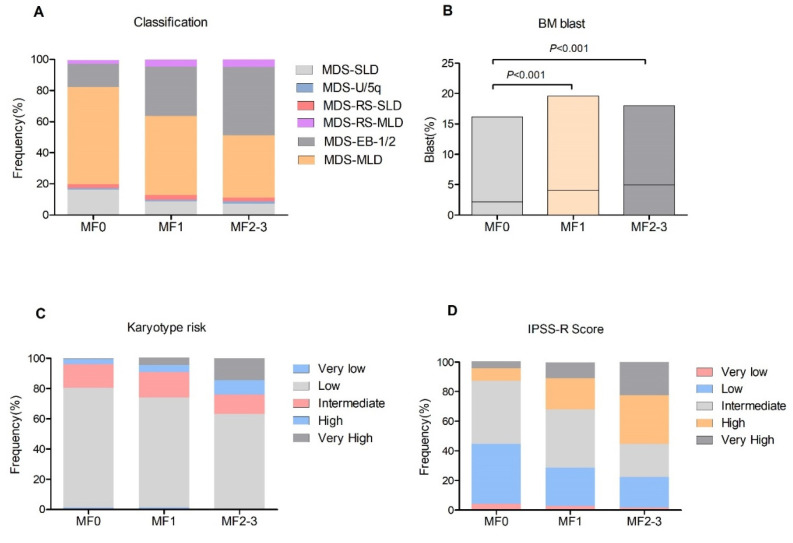
Clinical features of patients with BM fibrosis. (**A**), disease classification among different grade of fibrosis; (**B**), BM blast levels; (**C**), karyotype distribution according to IPSS-R; (**D**), IPSS-R scores.

**Figure 2 cancers-14-02984-f002:**
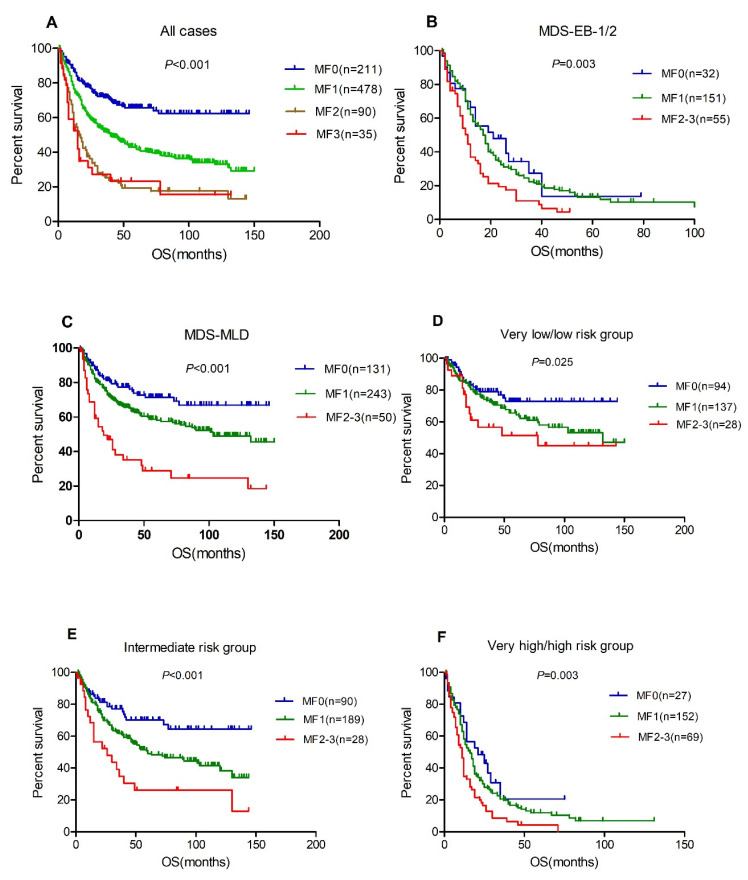
BM fibrosis associated with inferior overall survival. (**A**), total patients, *n* = 814; (**B**), MDS-EB-1 and 2, *n* = 238; (**C**), MDS-MLD, *n* = 424; (**D**), very low/low risk group; (**E**), intermediate risk group; (**F**), very high/high risk group.

**Figure 3 cancers-14-02984-f003:**
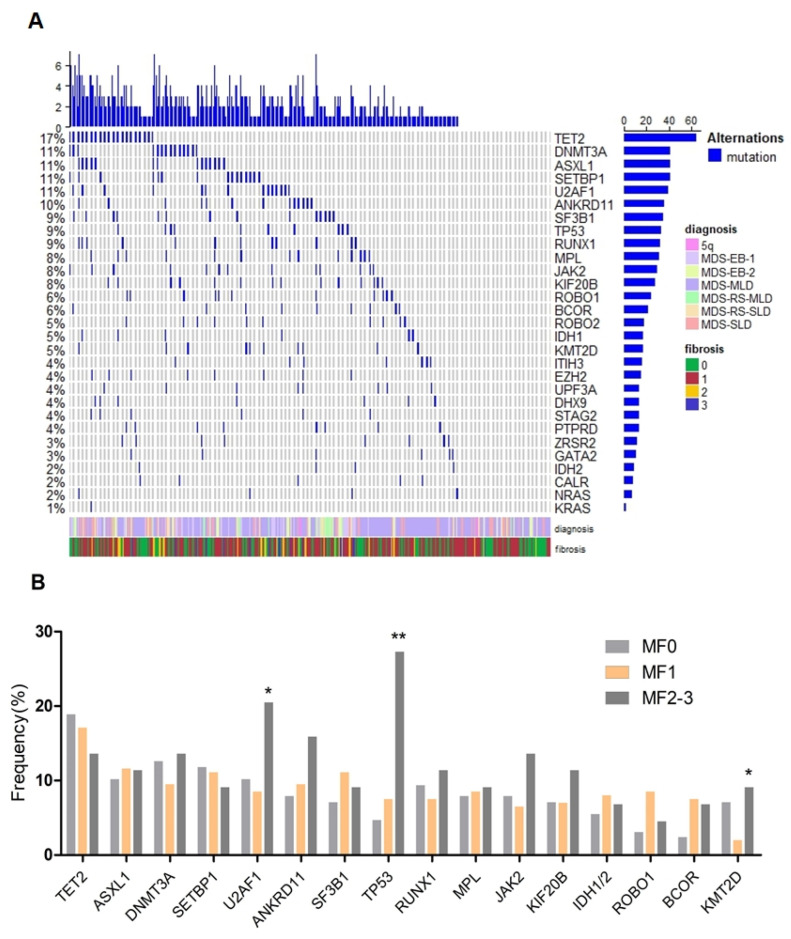
Mutational landscape of MDS in the cohort. (**A**), heat map showing the somatic mutational profile detected on next-generation sequencing; (**B**), frequency of mutations in different fibrosis subtypes, which are shown in indicated colors. *: *p* < 0.05; **: *p* < 0.001.

**Figure 4 cancers-14-02984-f004:**
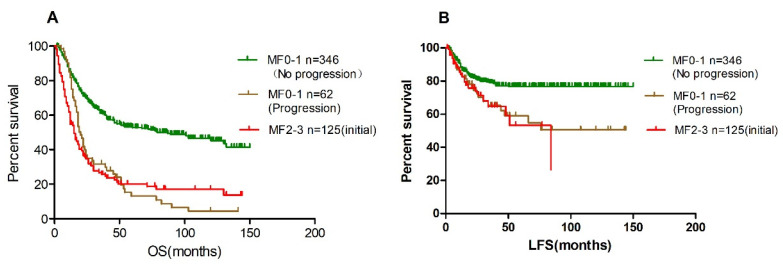
Patients with progressed fibrosis had poor overall survival and leukemia-free survival. (**A**), overall survival; (**B**), leukemia-free survival.

**Table 1 cancers-14-02984-t001:** Patient characteristics.

Variable	Total N = 814	MF0 N = 211	MF1 N = 478	MF(2, 3) N = 125	*p* Value
**Median age**	59	58	59	63	0.039
**Sex**					
Male	471	117 (24.8)	272 (57.8)	82 (17.4)	
Female	343	94 (27.4)	206 (60.1)	43 (12.5)	
**MDS classification**					<0.001
MDS-SLD	85	34 (16.1)	42 (8.8)	9 (7.2)	
MDS-MLD	424	131 (62.1)	243 (50.8)	50 (40.0)	
MDS-RS-SLD	24	6 (2.8)	15 (3.1)	3 (2.4)	
MDS-RS-MLD	33	5 (2.4)	22 (4.6)	6 (4.8)	
RA-EB-1/2	238	32 (15.2)	151 (31.6)	55 (44.0)	
MDS-U/5q	10	3 (1.0)	5 (1.0)	2 (1.6)	
**Cytopenia**					
Hemoglobin (g/L)	78.9	82.1	79.2	72.6	0.001
WBC counts					
(×10^9^/L)	3.8	4.0	3.6	4.2	0.204
Platelet counts					
(×10^9^/L)	86.8	83.3	86.2	94.8	0.592
**BM blast**					<0.001
<5%	581	180 (85.3)	329 (68.8)	72 (57.6)	
≥5%	233	31 (14.7)	149 (31.2)	53 (42.4)	
**Karyotype**					<0.001
Very good	8	2 (1.0)	6 (1.3)	0 (0)	
Good	595	168 (79.6)	348 (72.8)	79 (63.2)	
Intermediate	130	33 (15.6)	81 (16.9)	16 (12.8)	
Poor	41	7 (3.3)	22 (4.6)	12 (9.6)	
Very poor	40	1 (0.5)	21 (4.4)	18 (14.4)	
**IPSS-R**					<0.001
Very low	24	9 (4.3)	13 (2.7)	2 (1.6)	
Low	235	85 (40.3)	124 (25.9)	26 (20.8)	
Intermediate	307	90 (42.7)	189 (39.5)	28 (22.4)	
High	159	18 (8.5)	100 (20.9)	41 (32.8)	
Very high	89	9 (4.3)	52 (10.9)	28 (22.4)	

**Table 2 cancers-14-02984-t002:** Associations of mutations and BM fibrosis.

Mutation	Total N = 370	MF0 (*n*/%) N = 127	MF1 (*n*/%) N = 199	MF2-3 (*n*/%) N = 44	*p* Value
TET2	64 (17.3)	24 (18.9)	34 (17.1)	6 (13.6)	0.755
ASXL1	41 (11.1)	13 (10.2)	23 (11.6)	5 (11.4)	0.944
DNMT3A	41 (11.1)	16 (12.6)	19 (9.5)	6 (13.6)	0.542
SETBP1	41 (11.1)	15 (11.8)	22 (11.1)	4 (9.1)	0.917
U2AF1	39 (10.5)	13 (10.2)	17 (8.5)	9 (20.5)	0.079
ANKRD11	36 (9.7)	10 (7.9)	19 (9.5)	7 (15.9)	0.294
SF3B1	35 (9.5)	9 (7.1)	22 (11.1)	4 (9.1)	0.498
TP53	33 (8.9)	6 (4.7)	15 (7.5)	12 (27.3)	<0.001
RUNX1	32 (8.6)	12 (9.4)	15 (7.5)	5 (11.4)	0.581
MPL	31 (8.4)	10 (7.9)	17 (8.5)	4 (9.1)	0.964
JAKA2	29 (7.8)	10 (7.9)	13 (6.5)	6 (13.6)	0.269
KIF20B	28 (7.6)	9 (7.1)	14 (7.0)	5 (11.4)	0.564
IDH1/2	26 (7.0)	7 (5.5)	16 (8.0)	3 (6.8)	0.705
ROBO1	23 (6.2)	4 (3.1)	17 (8.5)	2 (4.5)	0.142
BCOR	21 (5.7)	3 (2.4)	15 (7.5)	3 (6.8)	0.102
ROBO2	18 (4.9)	4 (3.1)	12 (6.0)	2 (4.5)	0.502
KMT2D	17 (4.6)	9 (7.1)	4 (2.0)	4 (9.1)	0.022
ITIH3	16 (4.3)	6 (4.7)	8 (4.0)	2 (4.5)	0.936
EZH2	15 (4.1)	4 (3.1)	7 (3.5)	4 (9.1)	0.221
UPF3A	13 (3.5)	5 (3.9)	7 (3.5)	1 (2.3)	1.000
PTPRD	13 (3.5)	4 (3.1)	6 (3.0)	3 (6.8)	0.401
STAG2	13 (3.5)	2 (1.6)	8 (4.0)	3 (6.8)	0.183
DHX9	13 (3.5)	4 (3.1)	6 (3.0)	3 (6.8)	0.401
ZRSR2	12 (3.2)	5 (3.9)	7 (3.5)	0 (0)	0.538
GATA2	11 (3.0)	5 (3.9)	5 (2.5)	1 (2.3)	0.821
NRAS/KRAS	9 (2.4)	1 (0.8)	7 (3.5)	1 (2.3)	/
CALR	8 (2.2)	2 (1.6)	5 (2.5)	1 (2.3)	/

**Table 3 cancers-14-02984-t003:** The characteristics of patients with progressed MF.

Variable	Total N = 408	MF0-1 (Stable) N = 346	MF0-1 (Progression) N = 62	*p* Value
**Median age**	58	56	60	0.038
**MDS classification**				0.068
MDS-SLD/5q/MDS-U	35	29 (8.4)	6 (9.7)	
MDS-RS-SLD/MLD	26	21 (6.1)	5 (8.1)	
MDS-MLD	222	198 (57.2)	24 (38.7)	
MDS-EB-1/2	125	98 (28.3)	27 (43.5)	
**Cytopenia**				
Hemoglobin (mean g/L)	57.5	58.5	51.8	0.073
WBC counts				
(mean × 10 **^9^**/L)	3.6	3.4	4.4	0.061
Platelet counts				
(mean × 10 **^9^**/L)	75.8	72.2	96.0	0.03
**Blast**	3.9	3.6	5.1	<0.001
**Karyotype**				0.002
Very good/Good	303	263 (76.0)	40 (64.5)	
Intermediate	71	62 (17.9)	9 (14.5)	
Poor	19	13 (3.8)	6 (9.7)	
Very poor	15	8 (2.3)	7 (11.3)	
**IPSS-R**				0.04
Very low/Low	124	110 (31.8)	14 (22.6)	
Intermediate	161	139 (40.2)	22 (35.5)	
High	82	68 (19.7)	14 (22.6)	
Very high	41	29 (8.4)	12 (19.4)	

**Table 4 cancers-14-02984-t004:** Associations of mutations and BM fibrosis progression.

Mutation	Total N = 170	MF0-1 (Stable) N = 142	MF0-1 (Progression) N = 28	*p* Value
TET2	28 (16.5)	21 (14.8)	7 (25.0)	0.262
DNMT3A	22 (12.9)	20 (14.1)	2 (7.1)	0.537
SETBP1	18 (10.6)	15 (10.6)	3 (10.7)	1.000
ASXL1	17 (10.0)	13 (9.2)	4 (14.3)	0.487
ROBO1	16 (9.4)	16 (11.3)	0 (0)	0.077
SF3B1	14 (8.2)	10 (7.0)	4 (14.3)	0.252
RUNX1	14 (8.2)	9 (5.3)	5 (17.9)	0.058
JAKA2	14 (8.2)	10 (7.0)	4 (14.3)	0.252
KIF20B	14 (8.2)	12 (8.5)	2 (7.1)	1.000
U2AF1	13 (7.6)	10 (7.0)	3 (10.7)	0.452
ANKRD11	13 (7.6)	10 (7.0)	3 (10.7)	0.452
MPL	13 (7.6)	11 (7.7)	2 (7.1)	1.000
IDH1/2	13 (7.6)	11 (7.7)	2 (7.1)	1.000
BCOR	13 (7.6)	12 (8.5)	1 (3.6)	0.696
ITIH3	9 (5.3)	8 (5.6)	1 (3.6)	1.000
TP53	8 (4.7)	3 (2.1)	5 (17.9)	0.003
ROBO2	8 (4.7)	8 (5.6)	0 (0)	0.356
ZRSR2	8 (4.7)	6 (4.2)	2 (7.1)	0.620
PTPRD	7 (4.1)	7 (4.9)	0 (0)	0.601
UPF3A	6 (3.5)	5 (3.5)	1 (3.6)	1.000
DHX9	6 (3.5)	5 (3.5)	1 (3.6)	1.000
GATA2	6 (3.5)	4 (2.8)	2 (7.1)	/
EZH2	5 (2.9)	3 (2.1)	2 (7.1)	/
STAG2	4 (2.4)	4 (2.8)	0 (0)	/
KMT2D	2 (1.2)	2 (1.4)	0 (0)	/
NRAS/KRAS	2 (1.2)	1 (0.7)	1 (3.6)	/
CALR	2 (1.2)	2 (1.4)	0 (0)	/

## Data Availability

The datasets generated during the current study are available from the corresponding author on reasonable request.

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
