# Peer review of "Bone Marrow Fibrosis at Diagnosis and during the Course of Disease Is Associated with TP53 Mutations and Adverse Prognosis in Primary Myelodysplastic Syndrome"

_cancers, 2022, doi:10.3390/cancers14122984_

Round 1

Reviewer 1 Report

The paper try to understand the impact of marrow fibrosis in the prognosis of MDS.

- Bone marrow fibrosis has already demonstrated to have an adverse impact in the prognosis of MDS. However, to date we haven’t been able to include in the international prognosis systems. The paper is a good example on the impact of fibrosis in OS, however I miss an attempt to propose a new prognostic classification.

- Another big comment, is that the current WHO classification for MDS is the one published in 2017, so I will appreciate to use that classification instead of the one from 2008

- In the other hand, one of the biggest challenges here is the external validity of the results, since in many cases fibrosis is not easy to measure. Probably this results need to be confirmed in the hands of other pathologist.

- It is interesting the association with TP53, U2AF1 and KMT2D, however, as we are moving forward to the molecular IPSS, it will be great to address if the adverse impact of bone marrow fibrosis is independent or not from the molecular profile.

- Finally, I suggest a deep review in the English.

Thank you!

Reviewer 2 Report

In their manuscript, Zhao and colleagues from Shanghai retrospectively evaluate the finding of fibrosis of the marrow in the setting of myelodysplasia, trying to make the case that fibrosis confers independent adverse prognosis in the setting of MDS.  The population under study consisted of 814 patients with a diagnosis of MDS over a ten year period at the authors' primary institution.  Were these patients consecutive?  Were the patients reviewed from another institution?  Did any come with a prior bone marrow boipsy. The authors state that "all cases had an appreciable bone marrow biopsy before treatment" but what does that mean?  Were all patients treated?  Weren't some patient's merely observed? Again, were the marrow aspirates and biopsy procured at the time of diagnosis.  The bone marrow fibrosis was graded, but was this done by a limited number of hematopathologists, blinded to the clinical features of the presentation?  Was there independent review?  Of the 814 patients with the diagnosis of MDS, how many had analysis of karyotype and FISH from the bone marrow?  Unfortunately, only 370 patients had sequencing done using a 29-gene panel.  This is likely to have an impact on the analysis of risk. The authors refer to "a normal reticulum" but this reviewer does not know what that means.  Presumable, there was no fibrosis.  The authors then analyzed clinical characteristics of patients with mild or no fibrosis with those who had grade 2/3 fibrosis.  They then compare a histologic diagnosis of RAEB1/2 in groups separated by no fibrosis, mild fibrosis, and intermediate/ severe fibrosis.  Why were groups MF0 and MF1 not pooled as they had been in the prior comparison of age?  The same split of MF0 and others were used to compare blast count and poor/ very poor karyotype.  Again, for how many was no karyotype done?  Were there patients who had extensive marrow fibrosis who had favorable or intermediate cytogenetics? 

The authors evaluated the finding of fibrosis as it impacted overall survival.  Were all patients treated similarly? If not, how does one account for differences in treatment and the effect of treatment on survival?  Interestingly, the rate of transformation to acute leukemia was about the same regardless of severity of fibrosis, but certainly more common than among those with no fibrosis.  Again, looking at survival, the MF0 and 1 groups are pooled and compared to those with MF2 or 3.  Again, it might make more sense to be uniform throughout section 3.2.

Among the minority of patients in whom gene sequencing was done, it is not surprising that certain high-risk mutations were more often seen in patients with more fibrosis.  Since the authors had removed patients with known myeloproliferative disease, typical mutations in JAK2 and CALR were rarely detected in this sample population.

The manuscript includes many figures, particularly Kaplan-Meier actuarial statistical figures, supporting the impact of fibrosis on survival.  Again, how did treatment differ among the different groups?  Why would staging be more affected by the degree of fibrosis than current molecular pathology?  How reproducible is the histopathologic diagnosis of fibrosis?
